# Computing the effects of excitatory-inhibitory balance on neuronal input-output properties

**Alex D. Reyes** [ID]*

Center for Neural Science, New York University, New York, New York, United States of America

* ar65@nyu.edu

## Abstract

In sensory systems, stimuli are represented through the diverse firing responses and receptive fields of neurons. These features emerge from the interaction between excitatory ($E$) and inhibitory ($I$) neuron populations within the network. Changes in sensory inputs alter this balance, leading to shifts in firing patterns and the input-output properties of individual neurons and the network. Although these phenomena have been extensively investigated experimentally and theoretically, the principles governing how $E$ and $I$ inputs are integrated remain unclear. Here, probabilistic rules are derived to describe how neurons in feedforward inhibitory circuits combine these inputs to generate stimulus-evoked responses. This simple model is broadly applicable, capturing a wide range of response features that would otherwise require multiple separate models, and offers insights into the cellular and network mechanisms influencing the input-output properties of neurons, gain modulation, and the emergence of diverse temporal firing patterns.

## Author summary

Sensory stimuli activate networks of excitatory and inhibitory neurons whose interactions shape how the brain represents information. An individual neuron's response therefore depends not only on the strength of excitatory input, but also on how inhibition is recruited as stimulus conditions change. These interactions alter firing thresholds, response gain, and temporal firing patterns, yet the principles governing how excitatory and inhibitory inputs combine remain unclear. In this study, I develop a simple probabilistic framework to describe how excitatory and inhibitory synaptic inputs interact in feedforward inhibitory circuits. I express neuronal input–output relationships in terms of the probability that excitation survives coincident inhibition, thereby linking firing responses directly to identifiable synaptic and network parameters. Using this framework, I show that the model accounts for key features observed in sensory systems, including multiplicative and additive gain modulation, non-monotonic input–output curves, and diverse temporal firing patterns evoked by brief or sustained stimuli. By unifying these

**Data availability statement:** The source code and data used to produce the results and analyses presented in this manuscript are available on a Github repository at https://github.com/AlexDReyes/ReyesPlosCompBio2025.git.

**Funding:** A.R. was supported by grant 1 R01 MH129031-01 National Institutes of Health https://www.nih.gov/. The funders had no role in study design, data collection and analysis, decision to publish, or preparation of the manuscript.

**Competing interests:** The author has declared that no competing interests exist.

phenomena within a single, analytically tractable description, I provide insight into how changes in excitatory–inhibitory balance flexibly regulate neuronal responses across sensory conditions and behavioral states.

## Introduction

A common motif in sensory systems is the feedforward inhibitory circuit, where excitatory afferents from an external source synapse onto both excitatory ($E$) and inhibitory ($I$) neurons, with inhibitory neurons then synapsing back onto $E$ cells [1–5]. During stimulation, interactions within this circuit generate complex dynamics and shape the receptive field properties of neurons. As stimulus parameters vary, the balance between $E$ and $I$ inputs shifts, leading to both qualitative and quantitative changes in neuronal sensitivity and evoked firing patterns [1,4,6,7].

A neuron's response to a stimulus is characterized by its input–output (I–O) curve, where the input typically refers to synaptic current and the output to firing rate. Compared to neurons *in vitro* [8,9], neurons *in vivo* exhibit more diverse I–O profiles. Background excitation and inhibition from ongoing network activity introduce membrane potential fluctuations, allowing neurons to respond even to weak inputs that would otherwise remain subthreshold, and to generate smoothly increasing I–O curves [10–13]. In some cases, responses initially increase with stimulus intensity but then decline after reaching a peak [14–17], likely due to strong inhibitory recruitment [18,19]. I–O curves also depend on stimulus duration: brief stimuli (5–50 ms) evoke responses sensitive to the timing of excitation and inhibition [2,4,20,21], while longer stimuli (hundreds of milliseconds) evoke responses that vary with the average synaptic current generated during barrages [22–24].

Moreover, the I–O curves may change depending on the animal's state, such as when it is at rest, in motion [25,26], or actively attending [27,28]. To maintain selectivity to stimulus features across states, the slope of the I–O curve should change without affecting the minimum input needed to evoke firing [13,29]. This multiplicative (or divisive) gain modulation can arise from feedback from neighboring $E$ and/or $I$ neurons [30–34], feedforward inhibition [21], or the combination of synaptic noise and conductance [10]. Additive (or subtractive) modulation, by contrast, shifts the activation threshold without altering gain, thereby affecting tuning curve width [13].

Finally, differences in $E$–$I$ balance can produce diverse temporal firing responses. Some neurons exhibit continuous firing throughout the stimulus duration, while others fire transiently at the stimulus onset [16,22,35–39] and/or at the offset [39–42]. These firing profiles are observed in cortical and subcortical neurons [42] and may be generated locally [22,41–44] or inherited from upstream sources [40]. Additionally, a neuron's response type may change based on the stimulus intensity [16] or whether the preferred stimulus is presented [35,38,45,46].

Identifying common operating principles across these phenomena will provide valuable insights into potential mechanisms. This study aims to derive rules for calculating $E$–$I$ balance in feedforward inhibitory circuits. The model combines $E$ and $I$

inputs probabilistically and links the associated changes in responses and I–O curves to the synaptic and network properties. The model reproduces and clarifies the conditions for gain modulation, non-monotonic I–O curves, and diverse firing patterns.

## Results

The following sections begin with an idealized feedforward inhibitory network to introduce core concepts of the probabilistic interaction between excitatory and inhibitory inputs. These principles are then extended to more physiologically realistic conditions that include multiple, temporally distributed inputs. Finally, the model is applied to examine neuronal input–output properties, such as gain modulation and temporal firing profiles.

### Simple model

Consider a hypothetical circuit consisting of a postsynaptic excitatory neuron (henceforth termed the "reference cell") and a single inhibitory neuron, both receiving an excitatory postsynaptic potential (EPSP) from a common external afferent (Fig 1A). During stimulation, the afferent evokes an EPSP in both cells with probability $p_E$, where $p_E$ denotes the probability that an EPSP occurs in a given target cell.

The firing of the $I$ cell—and consequently the occurrence of an inhibitory postsynaptic potential (IPSP) in the reference cell—requires that an EPSP first arise in the $I$ neuron. Conditional on the occurrence of an EPSP in the $I$ neuron, that neuron fires with probability

$$p_I = \Pr(\text{fire} \mid \text{EPSP in } I).$$

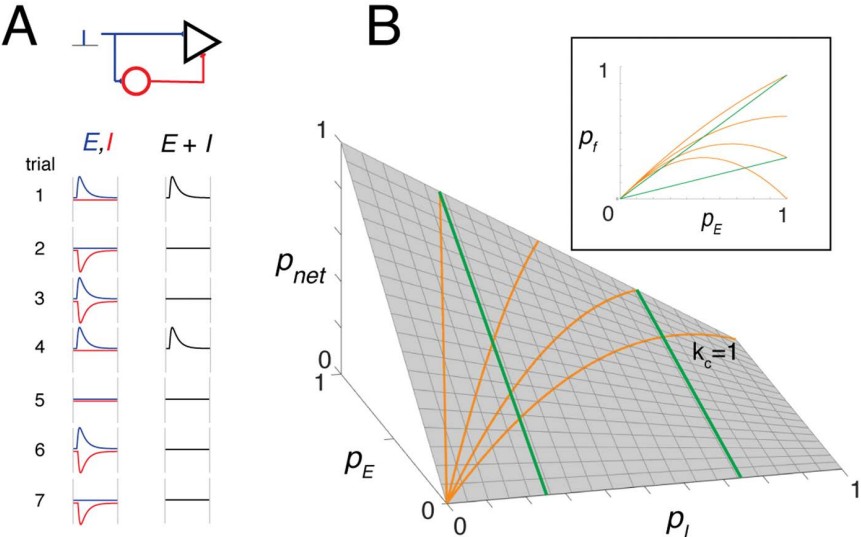

**Fig 1. Toy model illustrating the basic principles. A, top**, schematic of a simple feedforward inhibitory circuit. A postsynaptic excitatory neuron (black triangle) and a local inhibitory interneuron (red circle) both receive a single excitatory input from an external afferent. An IPSP is generated in the postsynaptic $E$ neuron whenever the interneuron fires. **Bottom**, EPSPs (blue) and IPSPs (red) recorded in the postsynaptic $E$ neuron over seven stimulus trials. An EPSP is shown as canceled when it coincides with an IPSP, but more generally Eq 1 describes the probabilistic outcome: over $n$ trials, the expected number of uncanceled EPSPs is $np_{net}$. **B**, surface plot of the net probability as a function of $p_E$ and $p_I$. Orange curves show predicted firing probability when increases linearly with ($p_I = k_c p_E$) at different scaling factors $k_c$. Green curves correspond to cases where is held constant. **Inset**, projection of these curves onto the $p_E - p_{net}$ plane. If each EPSP is suprathreshold, is equivalent to the firing probability $p_f$, and the plot can be interpreted as the neuron's input–output relation.

Thus, the joint probability that an EPSP appears in the $I$ neuron and that the $I$ neuron subsequently fires to generate an IPSP in the reference cell is $p_E p_I$.

Fig 1A shows four EPSPs (blue) and four IPSPs (red) across seven stimulus trials. For illustrative purposes, EPSPs and IPSPs are assumed to have the same amplitude and latency. An EPSP that coincides with an IPSP is effectively canceled (trials 3 and 6), whereas trials in which the IPSP fails to appear (trials 1 and 4) or appears alone (trials 2 and 7) have no effect. Thus, the probability that an EPSP is not canceled by an IPSP is

$$p_{\text{net}} = p_E - p_E p_I$$
$$= p_E(1 - p_I). \tag{1}$$

Eq 1 should be interpreted as a *trial-averaged survival probability*, reflecting both the chance that an excitatory input occurs ($p_E$) and the chance that it is not canceled by coincident inhibition ($1 - p_I$). Over $n$ independent trials, the number of uncanceled EPSPs in the reference cell follows a binomial distribution with mean $np_{\text{net}}$.

The surface plot in Fig 1B illustrates how $p_{\text{net}}$ varies with $p_E$ and $p_I$, highlighting key properties of Eq 1. First, $p_{\text{net}}$ is nonzero even when $p_E = p_I$, except in the limiting cases $p_E = 0$ or $p_I = 1$. Second, $p_{\text{net}}$ depends on how $p_E$ and $p_I$ covary. For example, if $p_I$ scales with $p_E$ as $p_I = k_c p_E$ with $0 \leq k_c \leq 1$, then $p_{\text{net}}$ increases monotonically with $p_E$ for small $k_c$, but becomes non-monotonic for larger $k_c$ (orange curves). Third, if $p_I$ is fixed, $p_{\text{net}}$ increases linearly with $p_E$ at a rate given by the slope $(1 - p_I)$ (green curves).

The inset shows the orange and green curves projected onto the $p_{\text{net}}$–$p_E$ plane. If each EPSP is suprathreshold, $p_{\text{net}}$ corresponds to the firing probability $p_f$, and the resulting plot represents the I–O relation of the neuron, with $p_E$ serving as a proxy for stimulus intensity or feature. This framework thus links the probabilistic structure of synaptic inputs to the macroscopic I–O behavior of the neuron.

In the following sections, this toy model is extended to more realistic networks with multiple afferents and inhibitory neurons, where cancellation emerges statistically.

## General model

Neuronal responses depend in part on the duration of stimulation. During sustained input, afferents and inhibitory neurons generate sequences of excitatory and inhibitory synaptic inputs, respectively, producing a synaptic barrage in the postsynaptic reference neuron. If the input is sufficiently strong, neurons fire repetitively at a rate determined by the average synaptic current [23]. In contrast, brief stimuli evoke EPSPs and IPSPs that arrive in close temporal proximity, placing the neuron in a regime where spiking is highly sensitive to both the amplitude and timing of synaptic inputs [4]. These two regimes will be treated separately, as the model yields distinct predictions for each. The case with sustained stimulation is examined first.

**Sustained stimuli: Oscillatory firing regime.** The response of the reference cell to long-duration stimuli was evaluated in three stages (Fig 2A). First, Poisson-distributed spike trains from $n_E$ external afferents (blue) were generated to produce excitatory synaptic barrages, which were delivered to both the reference cell and the $I$ neurons, modeled as leaky integrate-and-fire (LIF) units. Second, the spike trains evoked in a specified number of $I$ neurons (red) were summed to form the inhibitory synaptic barrage. Finally, the excitatory and inhibitory barrages were combined and delivered to the reference cell, and its firing response was measured across a range of conditions.

To introduce the key variables manipulated in the simulations, the relationships between the $E$ and $I$ probabilities and the corresponding synaptic currents are first developed. A detailed description of the model equations and parameters is provided in the Methods and S1 Appendix; here, only the principal features necessary for interpreting the Results are summarized.

Fig 2A shows a schematic of the feedforward network, with the parameters defined in Table 1. During a prolonged stimulus, each afferent generated a Poisson train of action potentials with mean rate $p_E p_{E \rightarrow R} r_E$, where $p_E$ is the probability

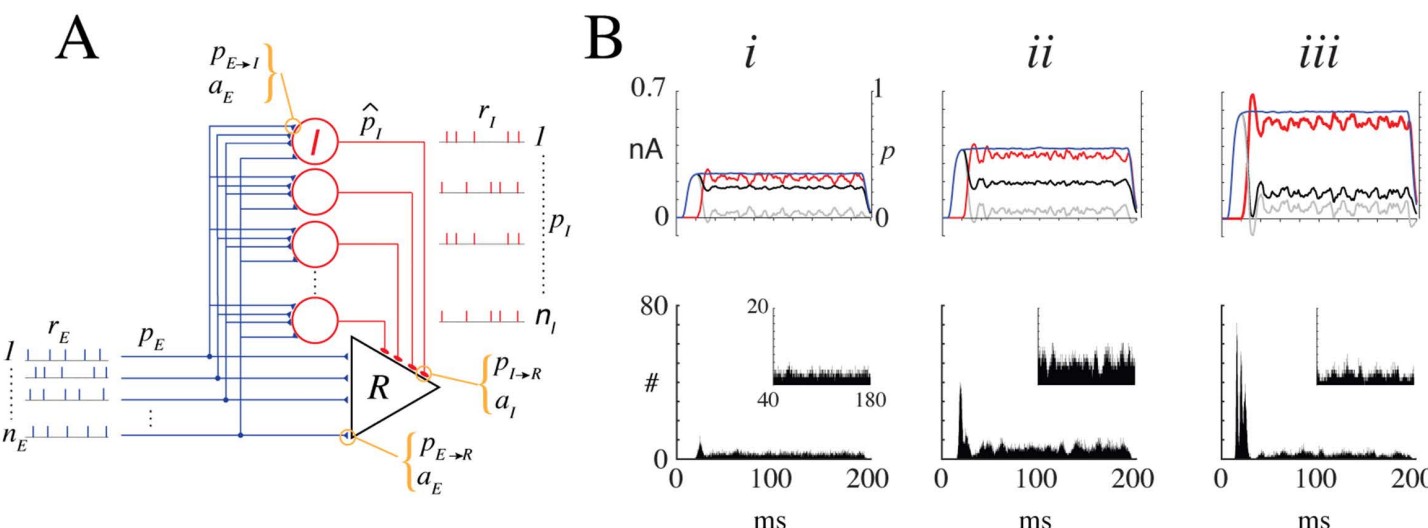

**Fig 2. Predicting firing rate in response to sustained stimuli. A**, Schematic of the network consisting of a reference cell (R, triangle) and $n_I$ inhibitory neurons (red circles). During stimulation, each of the $n_E$ afferents fires with probability $p_E$ at rate $r_E$ for a duration of 0.1 to 1 second. Each afferent spike evokes an EPSC with amplitude $a_E$ in both the reference neuron and each inhibitory neuron with probabilities $p_{E\to R}$ and $p_{E\to I}$, respectively. If the excitatory synaptic barrage was sufficiently large, each inhibitory neuron fired with probability $\hat{p}_I$ at a rate $r_I$. The effective probability—which factors in network variables, synaptic strength, and firing rates—that inhibitory inputs reach the reference cell is $p_I$ (see text). Finally, the probability that each inhibitory spike evoked an inhibitory postsynaptic current (IPSC) with amplitude $a_I$ in the reference neuron is $p_{I\to R}$. In the simulations, $p_{E\to R}$ and $p_{I\to R}$ are set to 1. **B**$i$, top, Time course of excitatory synaptic current barrage (blue, left ordinate) when $p_E$ (right ordinate) was ramped to a steady-state value of 0.35. The inhibitory barrage (red), plotted as an absolute value for comparison, develops after a short delay of $\sim$ 20 ms, eventually reaching a comparable steady-state value. The resulting net current predicted by the probabilistic model (black), $p_{\text{net}} = p_E(1-p_I)$, shows a transient peak at stimulus onset followed by a lower steady-state level. In contrast, the unconditioned current obtained by direct subtraction of excitatory and inhibitory currents is nearly zero (gray). **Bottom**, Spike histogram of the reference neuron. **Inset**, magnified view of the tonic firing component. $ii, iii$, Same as $i$, except with $p_E$ and $p_I$ set to 0.55 and 0.85, respectively. Histograms compiled with bin width = 0.01 ms and with 10,000 sweeps. **Model parameters**, Simulations used leaky integrate-and-fire neurons (see Methods) with $n_E = 250$, $n_I = 50$, $r_E = 50$ Hz, and $r_I \approx 20$–100 Hz. EPSC and IPSC are alpha functions with amplitudes $a_E = 10.3$ and $a_I = -10.3$ **(pA)**. The steady-state value of $p_I$ was controlled by fixing the input probability to inhibitory cells ($\pi_{E\to I}$ in Eq 2) at 0.35 and adjusting the ratio $k_n$ (Eq 3) to 0.3 (**i**), 0.4 (**ii**), and 0.7 (**iii**). Bin width = 0.01 ms.

that an afferent becomes active during the stimulus, $p_{E\to R}$ is the probability that each spike produces an excitatory postsynaptic current (EPSC) in the reference cell, and $r_E$ is the firing rate conditional on activation. Across afferents or repeated trials, this product represents the effective EPSC rate, and the probability of observing an EPSC within a time bin of width $\Delta t$ is approximately $p_E \, p_{E\to R} \, r_E \, \Delta t$. In this formulation, $p_E$ captures variability across afferents or stimulus presentations, $p_{E\to R}$ quantifies synaptic efficacy, and $r_E$ describes the within-trial firing dynamics of an active afferent.

When the $n_E$ afferents fired, each spike evoked an EPSC, whose integral yielded the total charge transfer $q_E$. The mean steady-state excitatory currents to the reference neuron ($\bar{i}_{E\to R}$) and to the inhibitory neurons ($\bar{i}_{E\to I}$) are shown in Fig 2B (blue; see S1 Appendix). When the afferent input was sufficiently large, the $I$ neurons fired at a rate $r_I$ and generated inhibitory synaptic current ($\bar{i}_{I\to R}$) in the reference cell. These mean currents are expressed as

$$\bar{i}_{E\to R} = p_E \, p_{E\to R} \, n_E \, q_E \, r_E,$$
$$\bar{i}_{E\to I} = p_E \, p_{E\to I} \, n_E \, q_E \, r_E,$$
$$= \pi_{E\to I} \, n_E \, q_E \, r_E$$
$$\bar{i}_{I\to R} = \hat{p}_I \, p_{I\to R} \, n_I \, q_I \, r_I$$
$$= p_I \, p_{E\to R} \, n_E \, q_E \, r_E. \tag{2}$$

**Table 1. Parameters of the feedforward network.**

| Symbol | Description |
| --- | --- |
| $p_E$ | Prob. that a stimulus activates an afferent |
| $p_I$ | Effective Prob. that an inh. input reaches the reference cell |
| $p_{E \to R}$ | Prob. that an afferent spike evokes an *exc.* synaptic input in the reference cell |
| $p_{E \to I}$ | Prob. that an afferent spike evokes an *exc.* synaptic input in an *I* cell |
| $p_{I \to R}$ | Prob. that an *I* cell spike evokes an inh. synaptic input in the reference cell |
| $n_E, n_I$ | Numbers of *E* afferents and *I* neurons |
| $r_E, r_I$ | Firing rates of *E* afferents and *I* neurons |
| $a_E, a_I$ | Amplitudes of *E* and *I* postsynaptic currents |
| $q_E, q_I$ | Charge transfers of *E* and *I* synaptic events |

Here, $p_{E \to I}$ denotes the probability that an afferent spike evokes an EPSC in an inhibitory neuron, and $p_{I \to R}$ denotes the probability that an inhibitory spike evokes an IPSC in the reference cell [8,47–50]. For convenience, the effective excitatory drive to inhibitory neurons is defined as $\pi_{E \to I} = p_E\, p_{E \to I}$.

The term $p_I$ denotes the *effective* probability that an inhibitory input reaches the reference cell. It is expressed as a product of factors that relate inhibitory neuron number, synaptic strength, and activity level to those of the excitatory afferents (see S1 Appendix):

$$p_I = \hat{p}_I\, k_n\, k_q\, k_r\, k_{EI}, \tag{3}$$

where $\hat{p}_I$ is the probability that an inhibitory neuron fires during the stimulus, $k_n = \frac{n_I}{n_E}$ is the inhibitory-to-excitatory input ratio, $k_q = \left| \frac{q_I}{q_E} \right|$ is the ratio of unitary charge transfers, $k_r = \frac{r_I}{r_E}$ is the inhibitory-to-excitatory rate ratio, and $k_{EI} = \frac{p_{I \to R}}{p_{E \to R}}$ is the synaptic efficacy ratio. By construction, $p_I$ incorporates these relative differences in number, strength, and activity, allowing the inhibitory current (Eq 2) to be written in terms of $n_E$, $q_E$, $r_E$, and $p_{E \to R}$.

The mean net current to the reference cell, under the condition that inhibition is contingent on coincident excitation, is given by (see also Eq S12 in S1 Appendix)

$$\bar{I}_{\text{net}} = n_E\, q_E\, r_E\, p_{E \to R}\, p_{\text{net}},$$
$$\text{with} \quad p_{\text{net}} = p_E\,(1 - p_I), \tag{4}$$

where $p_{\text{net}}$ denotes the survival probability of excitation—requiring both that an excitatory input occurs with probability $p_E$ and that it is not canceled by coincident inhibition with probability $(1 - p_I)$ (see S1 Appendix for details). This compact probabilistic form incorporates interneuron recruitment and synaptic reliability, and provides an effective representation of the mean net excitatory drive under feedforward coupling. In contrast to classical add–subtract models (e.g., for Poisson processes [51]), in which excitation and inhibition are treated as independent and subtracted directly, the present formulation yields a probabilistic measure that, by construction, remains within a normalized range (approximately [0, 1]) for realistic values of identifiable parameters (see Discussion), thereby avoiding the unbounded subtraction of independent terms.

Fig 2B*i* shows the synaptic currents (top) and spike histograms (bottom) recorded from the reference cell during stimulation. Simulations were performed using LIF neurons (see figure caption and Methods for details). The afferent input probability was ramped over time to a steady-state value of $p_E = 0.35$, producing a barrage of excitatory synaptic currents (blue). Physiologically, this ramp mimics the gradual increase in stimulus intensity before reaching a plateau. When inhibition was treated as independent of coincident excitation, inhibitory currents (red) were recruited after a short delay and rose to a comparable level. In this case, the unconditioned net current—obtained by direct subtraction of excitatory and

inhibitory currents, $i_{E\to R}(t) - i_{I\to R}(t)$ (gray)—was nearly zero, and no firing occurred. By contrast, when inhibition was conditioned on coincident excitation, the net synaptic current (black, top) and the firing profile of the reference cell (bottom) displayed a small transient peak followed by a sustained tonic component.

The net current and tonic firing reflected the balance between $p_E$ and $p_I$, in agreement with the toy model (orange curves in Fig 1B). When both probabilities increased together, $\bar{i}_{net}$ and the tonic firing rate rose to a peak value (Fig 2B*ii*) before declining (*iii*), and eventually vanished as $p_E$ and $p_I$ approached 1.

Simulations were performed while systematically increasing $p_E$. The resulting input–output (I–O) curve was obtained by plotting the evoked firing rate (mean ± SD) against $p_E$ (Fig 3A). When the mean net input current $\bar{i}_{net}$ exceeded the rheobase $i_{rh}$, the reference neuron entered the oscillatory firing regime. In this regime, the average firing rate of the LIF model (green curve) followed the standard analytical solution (see Eq S16 of S1 Appendix).

Firing could still occur with weak inputs due to voltage fluctuations that occasionally crossed threshold, even when $\bar{i}_{net}$ was below rheobase (Fig 3A, inset). In this fluctuation-dominated (sub-oscillatory) regime (cyan curves), the firing rate was given by

$$r_{\text{fluct}} = \Pr(X \geq n_\theta)\, r_{rh}, \tag{5}$$

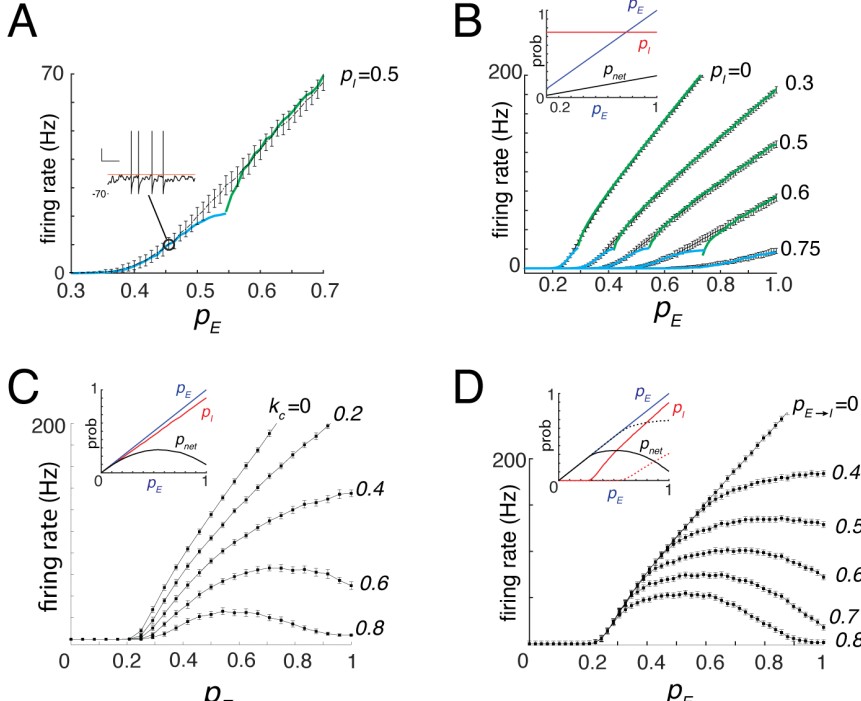

**Fig 3. Input–output curves and gain modulation. A**, Representative input–output (I–O) curve showing mean firing rate (mean ± SD) as a function of $p_E$, with $p_I = 0.46$ ($\pi_{E\to I} = 0.5$, $k_n = 0.2$, $k_r \approx 2.3$). Superimposed are the predicted firing rates in the sub-oscillatory (cyan) and oscillatory (green) regimes. **Inset**, Example membrane potential trace evoked by a sub-rheobase input. Fluctuations in the membrane potential cause threshold crossings (orange line). **B**, I–O curves for different fixed values of $p_I$. The ratio $k_r = \frac{r_I}{r_E}$ in Eq 3 was varied by fixing $\pi_{E\to I}$ at specified levels (range: 0–0.8), thereby modulating $r_I$. **Inset**, Plots of $p_E$, $p_I = 0.75$, and $p_{net}$ versus $p_E$. **C**, Same as **B**, except $p_I$ increased linearly with $p_E$, implemented by setting $k_n = k_c p_E$ while holding $\pi_{E\to I} = 0.4$, $k_q = 1$, and $k_r \approx 1$ (see text). **D**, Same as **C**, except $\pi_{E\to I}$ was allowed to vary with  for different values of $p_{E\to I}$. **Inset**, $p_I$ remained near zero for low $p_E$ and increased linearly with $p_E$ once threshold was crossed. Solid and dotted curves correspond to $\pi_{E\to I} = 0.7$ and 0.4, respectively.

where $r_{rh}$ is the minimum oscillatory firing rate at rheobase and $\Pr(X \geq n_\theta)$ is the probability that the excitatory drive exceeds threshold (derivation in S1 Appendix). The resulting $r_{\text{fluct}}$ increased sigmoidally with $p_E$ and approached a maximum near the onset of the oscillatory regime. Across the full range of $p_E$, the overall firing rate was taken as the larger of $r_{\text{fluct}}$ and $r_{\text{osc}}$, reflecting the transition from fluctuation-driven to oscillatory firing as the mean input current crosses rheobase.

The model predicts that, for a fixed $p_I$, the slope of the I–O curve scales in proportion to $1 - p_I$ (Fig 1B, green; Eq 1). Consistent with this result, increasing $p_I$ (see figure caption for details) caused the I–O curves to exhibit a reduction in slope. However, there was also a rightward shift, reflecting a higher activation threshold (Fig 3B). This condition, in which inhibition was held constant across the full range of $p_E$, is analogous to experiments in which inhibitory neurons are continuously activated optogenetically during sensory stimulation [34]. The slope decrease corresponds to multiplicative gain modulation, whereas the threshold shift reflects an additive effect of persistent inhibition, requiring stronger excitation to elicit firing. These effects were observed in both the oscillatory and fluctuation-dominated regimes and were captured by the analytical expressions for $r_{\text{osc}}$ (green) and $r_{\text{fluct}}$ (cyan). Together, these results demonstrate that inhibition modulates the I–O relationship through a combination of multiplicative and additive gain mechanisms [13,29].

The I–O curves exhibited either monotonic or non-monotonic increases with $p_E$, depending on how strongly inhibition co-varied with excitation (Fig 3C), consistent with model predictions (Fig 1B, orange curves). Co-variation of $p_I$ with $p_E$ was implemented by allowing $k_n$ to increase linearly with $p_E$ ($k_n = k_c p_E$, $k_c \in [0, 1]$), while keeping constant (inset). Physiologically, this corresponds to the progressive recruitment of inhibitory neurons with increasing stimulus intensity. For small $k_c$, firing rates increased monotonically with $p_E$ (Fig 3C), whereas $k_c$ approached 1, the I–O curves flattened and eventually became non-monotonic. Notably, the minimum $p_E$ required to evoke firing ($\sim 0.2$) remained unchanged, while the slope of the rising phase decreased. Thus, when $p_E$ was restricted to the range in which the I–O curve increased, gain modulation was effectively multiplicative.

Finally, the model predicts changes in the I–O curves of the reference cell under more physiological conditions, in which inhibition strengthens as excitation increases. In earlier simulations, $p_I$ was manipulated by fixing the drive to inhibitory neurons (Eq 2). Here, the effective drive $\pi_{E \to I}$ was allowed to increase with $p_E$, mimicking increased drive to inhibitory neurons with rising stimulus intensity (Fig 3D, inset). The slope of this recruitment was controlled by varying $p_{E \to I}$, which physiologically corresponds to changes in the efficacy of the afferent synapse onto inhibitory neurons. Increasing $p_{E \to I}$ shifted the $p_I$ curve leftward (dotted to solid red, inset), causing the I–O curves of the reference neuron to become progressively more non-monotonic (Fig 3D). The curves shared a common threshold and overlapped at low $p_E$ before diverging at higher values. Although these changes do not conform to classical forms of gain modulation, they can still generate multiplicative effects on tuned inputs (see below).

**Correction for conductance effects.** A key assumption in Eqs 2 and 4 is the linear summation of excitatory and inhibitory inputs. This assumption is violated when synaptic inputs alter the total membrane conductance, since the net current at rest can differ substantially from that during depolarized states, leading to errors in predicted firing rates. To account for this effect, a conductance-dependent adjustment was derived (see S1 Appendix). Incorporating this correction substantially improved prediction accuracy (Fig S2 of S1 Appendix) while preserving the probabilistic formulation, allowing the same framework to be applied to conductance-based synapses.

**Gain modulation of tuned responses.** The slope changes in the I–O curves described above (Fig 3B–3D) suggest mechanisms for multiplicative gain modulation of neural responses [27,28,52]. To test this, simulations were performed with $p_E$ following a Gaussian profile representing tuned sensory input (Fig 4A). Three conditions were examined: (mode 1) fixed $p_I$; (mode 2) $p_I$ increasing linearly with $p_E$; and (mode 3) $p_I$ following the I–O curve of the inhibitory neurons. In all cases, the peak input ($p_E = 0.35$) produced modulated tuning curves with peak firing rates within 50% of the control (60 Hz), consistent with experimental data [27,28,52].

For mode 1, relatively small values of $p_I$ were sufficient to produce multiplicative gain modulation. Although the I–O curves showed both slope and threshold changes (Fig 3B), the small range of $p_I$ that was used primarily affected the

slope (Fig 4B, *i*), resulting in a reduction in tuning curve amplitude (*ii*) with minimal change in width (*iii*). Physiologically, such low $p_I$ values (Eq 3) could arise from a small number of inhibitory neurons relative to afferents (low $k_n$), weak synaptic inhibition (low $k_q$), and/or low inhibitory firing rates (low $k_r$).

In mode 2, multiplicative gain modulation was achieved by setting the scalar $k_c$ in the relationship $k_n = k_c p_E$ to modest values. The resulting I–O curves were monotonic, with progressively reduced slopes as $k_c$ increased (Fig 4C, *i* and inset). The peaks of the corresponding tuned responses (*ii*) varied with $k_c$ and, when normalized, superimposed (*iii*). To achieve a linear relationship between $p_I$ and $p_E$ under physiological conditions would require a fixed $\pi_{E\rightarrow I}$ across input intensities and low inhibitory firing thresholds to ensure engagement even for small $p_E$ values.

In mode 3, multiplicative gain modulation occurred but only within a limited range of $p_E$. At low $p_E$, the I–O curves overlapped substantially (Fig 4D, *i*), and divergence required strong inhibition at moderate $p_E$, where slope differences emerged. With larger $p_E$, the I-O curves became non-monotonic (inset). When restricted to the rising phase, peak responses of the tuning curve could be modulated (*ii*) with minimal changes in width (*iii*). However, when $p_E$ exceeded the range of the rising phase, firing shifted to the decaying portion of the curve, producing a central dip and bimodal tuning (Fig S3 of S1 Appendix).

**Temporal firing profiles.** Neuronal firing patterns can encode distinct stimulus features or task-related components [36,53,54]. As the stimulus changes, the drive to the neuron—reflected in variations of $p_E$—also changes. To examine

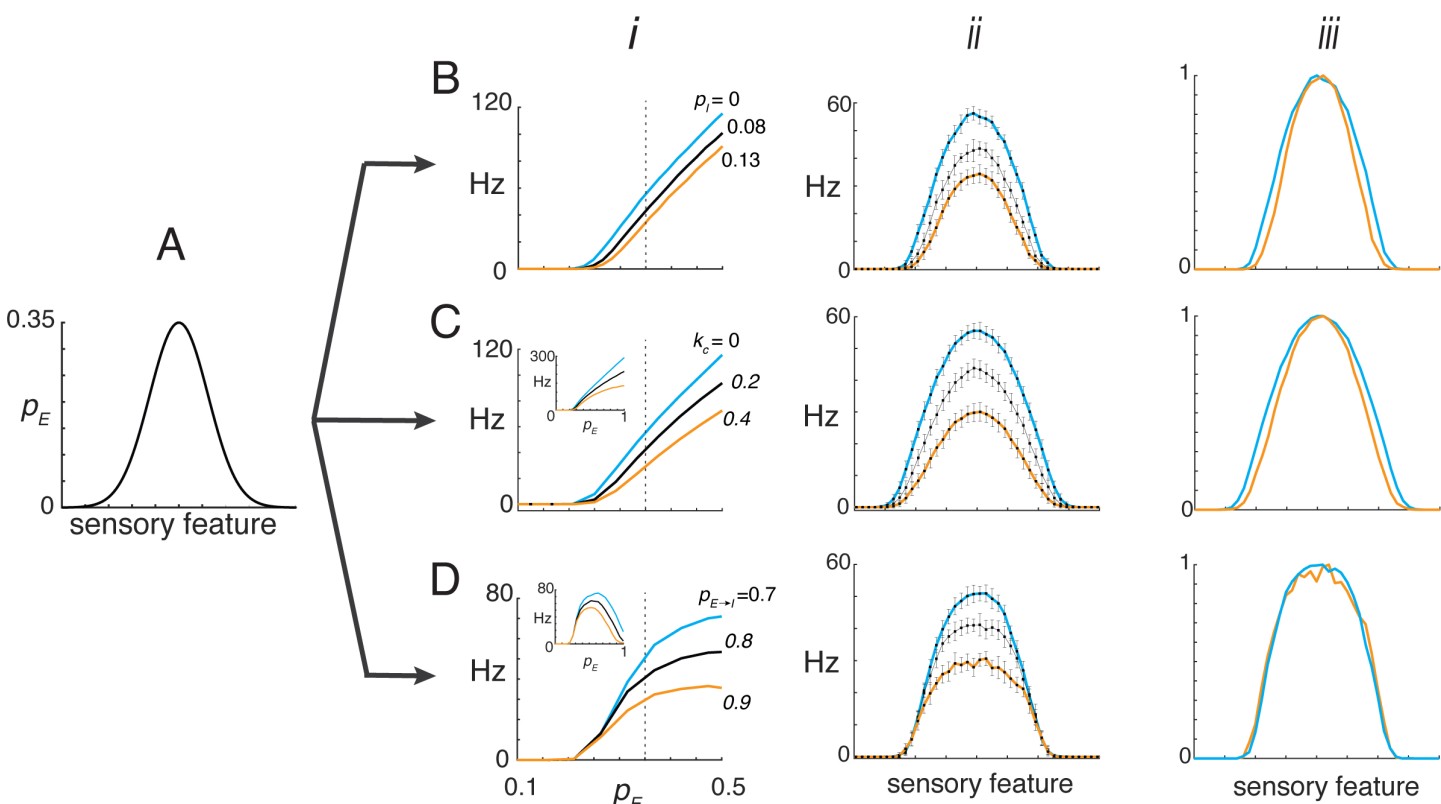

**Fig 4. Gain modulation of tuned responses. A**, Simulations in which $p_E$ followed a Gaussian profile representing a tuned sensory input. **B**, Case with constant $p_I$ set to 0, 0.08, or 0.13. *i*, I–O curves used to transform the input. *ii*, Resulting tuned responses of the reference neuron. *iii*, Tuned responses normalized to a peak value of 1. **C**, Same as in **B**, except $p_I$ increased linearly with $p_E$. This was implemented by setting $k_n = k_c\, p_E$ with $k_c = 0, 0.2,$ or $0.4$ at a fixed $\pi_{E\rightarrow I}$. **Inset**, Corresponding I–O curves across the full range of $p_E$. **D**, Same as in **C**, except $\pi_{E\rightarrow I}$ varied with $p_E$ at different rates by setting $p_{E\rightarrow I} = 0.7, 0.8,$ or $0.9$.

how such temporal profiles depend on E–I interactions, it is necessary to systematically vary the time-dependent inputs. Controlling the inhibitory drive is challenging, however, because its onset and magnitude depend on how inhibitory neurons are recruited by the stimulus (Fig 2B). This recruitment, in turn, depends on the biophysical properties of inhibitory neurons [55] and on the amplitude of their excitatory inputs [56–58]. The firing profile of the reference cell also depends on the relative timing of EPSPs and IPSPs: in most cases, IPSPs lag EPSPs [2,18,40], although they can also precede them under certain conditions [59,60].

Replicating the full range of possible time-varying relationships between $p_E(t)$ and $p_I(t)$ would require detailed modeling of inhibitory neuron dynamics, which is beyond the scope of this study. To allow direct and independent control of $p_I(t)$, the inhibitory neurons were bypassed, and inhibitory inputs were generated in the same way as the excitatory afferents. The excitatory input, together with the conditioned inhibition $p_E(t)p_I(t)$, was then delivered to the reference cell. This abstraction isolates the temporal interaction between excitation and inhibition, enabling their overlap to be examined in a controlled and systematic manner.

Both $p_E(t)$ and $p_I(t)$ were ramped up to the same steady-state value and then ramped down (blue and red dashed curves in Fig 5A–5C, bottom panels). The simulation parameters were identical (see Fig 5 captions and Methods for details) except that the relative onsets were varied. The barrages were calculated as above and delivered to the LIF neuron, and firing histograms were compiled across repeated trials (top panels).

A wide range of temporal firing patterns was generated by varying the magnitudes and relative timing (lags) of $p_E(t)$ and $p_I(t)$. In these simulations, the time courses of $p_E(t)$ and $p_I(t)$ were identical (Fig 5A–5C, bottom panels) but differed in their temporal lags (*i–iii*). For moderate inputs with no lag (A, *i*), the computed $p_{net}(t)$ (black) closely followed the time courses of $p_E(t)$ (blue) and $p_I(t)$ (red dashed), ramping to a steady level before decaying. The reference neuron responded with delayed tonic firing (top).

With larger input magnitudes (B), $p_{net}(t)$ exhibited transient peaks at stimulus onset and offset, accompanied by a reduced steady-state level. Further increases in steady-state amplitude (C) eliminated the tonic component entirely, leaving only onset and offset peaks, since $p_E(t)[1 − p_I(t)] \neq 0$ only during those intervals. Accordingly, the reference neuron fired exclusively at stimulus onset and offset (top).

When $p_I(t)$ lagged $p_E(t)$ by 2 ms (*ii*), similar results were obtained, except that the onset peak of $p_{net}(t)$ was larger than the offset peak (Fig 5C). This produced pronounced firing at stimulus onset—greater than that observed in the no-delay condition (*i*)—and no firing at stimulus offset. Conversely, when $p_I(t)$ preceded $p_E(t)$ (*iii*), spiking occurred only at stimulus offset.

**Brief stimuli: Transient firing regime.** Brief stimuli, such as tone pips in the auditory system [2,33,59], light flashes in the visual system [61], or whisker deflections in the somatosensory system [4,6], evoke a volley of compound EPSPs, followed by compound IPSPs from inhibitory neurons after a small delay. Whether the postsynaptic cell fires depends both on the relative magnitude and timing of these inputs, in contrast to long duration stimuli where firing is determined primarily by the average synaptic current.

To examine the effects of *E–I* balance and timing on firing probability, simulations were performed in the same network as above. A brief stimulus evoked EPSPs in both the reference and *I* neurons whose arrival times followed a Gaussian distribution (Fig 6A*i*, top panel, blue histogram). When the *I* neurons fired, they generated IPSPs in the reference cell a short time later, with a narrower temporal distribution (red). In response to these EPSPs and IPSPs, the reference cell fired action potentials with a narrower distribution (middle, gray) than the input, consistent with experimental observations [4].

The instantaneous excitatory and inhibitory probabilities (blue and red traces in Fig 6A, bottom panel) were computed from the compound EPSPs and IPSPs using convolution expressions derived in S1 Appendix. These are given by the time-dependent profiles

$$\text{Excitatory:} \quad p_E \tilde{p}_E(t),$$
$$\text{Inhibitory:} \quad p_I \tilde{p}_I(t), \tag{6}$$

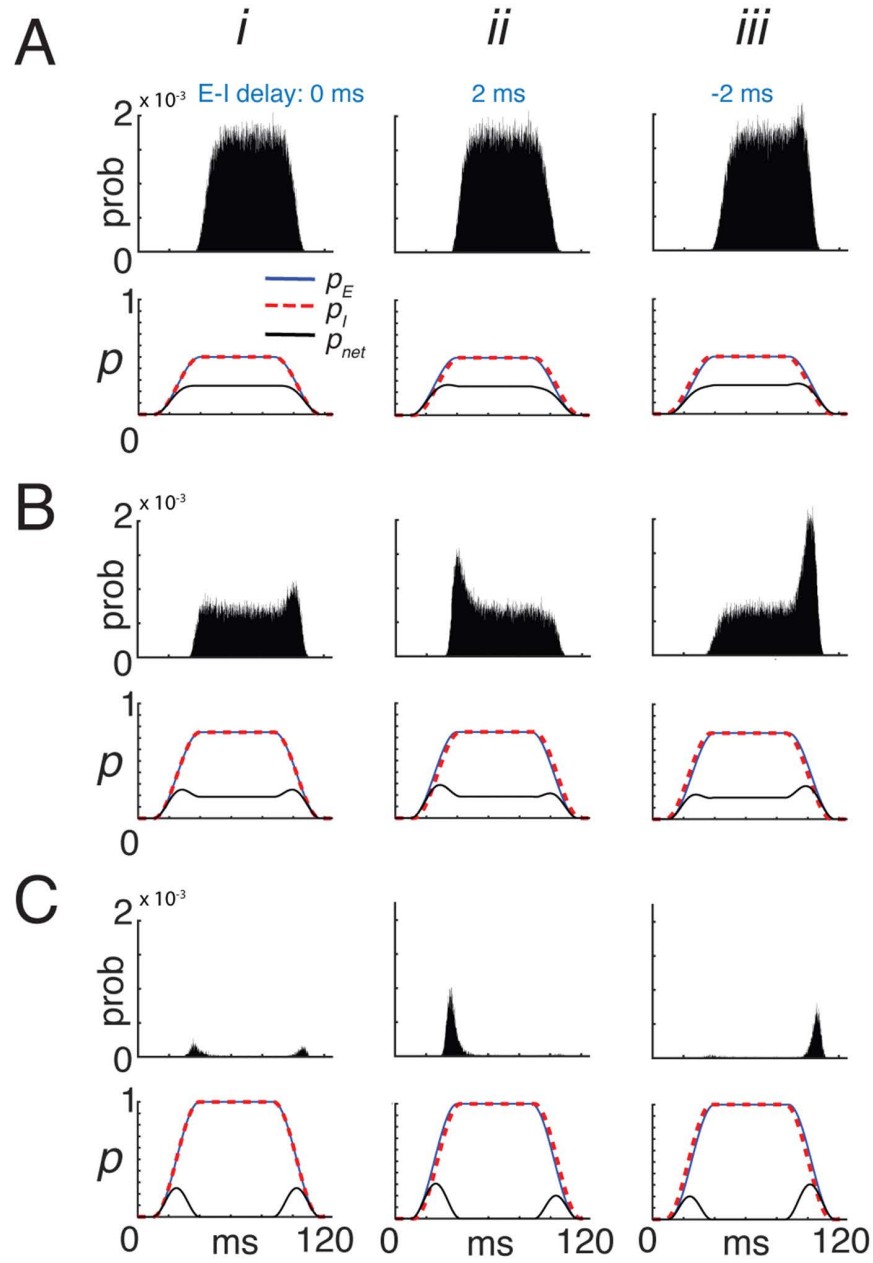

**Fig 5. Effects of inhibitory magnitude and timing on temporal firing. A*i*,** *bottom*: Excitatory and inhibitory input probabilities, $p_E(t)$ (blue) and $p_I(t)$ (red) rose and fell together with no delay, each reaching a steady-state value of 0.5. The computed $p_{net}(t) = p_E(t)\,[1 - p_I(t)]$ is superimposed (black). *Top*: the corresponding spike histogram shows tonic firing in the reference neuron. ***ii–iii***, Same as ***i***, except $p_I(t)$ lagged or led $p_E(t)$ by 2 ms, respectively. **B,** Same as **A**, but with larger steady-state probabilities (0.75). **C**, Same as **A**, but with steady-state probabilities equal to 1. **Model parameters:** LIF neuron as in Methods. Per-afferent input intensities were $\lambda_{E\to R}(t) = p_E(t)\,r_E\,n_E$ and $\lambda_{I\to R}(t) = p_E(t)\,p_I(t)\,r_I\,n_I$, with $r_E = r_I = 50$ Hz and population sizes $n_E = n_I = 250$, bin width = 0.01 ms.

where $\tilde{p}_E(t)$ is the excitatory arrival-time profile and $\tilde{p}_I(t)$ is the conditioned inhibitory arrival-time profile, both obtained by convolving peak-normalized unitary PSPs with the corresponding spike-time histograms. The effective probability $p_I$ is as defined above, except that the ratio of inhibitory to excitatory amplitudes $k_a = \left|\frac{a_I}{a_E}\right|$ replaces the ratio of charges $k_q$. The

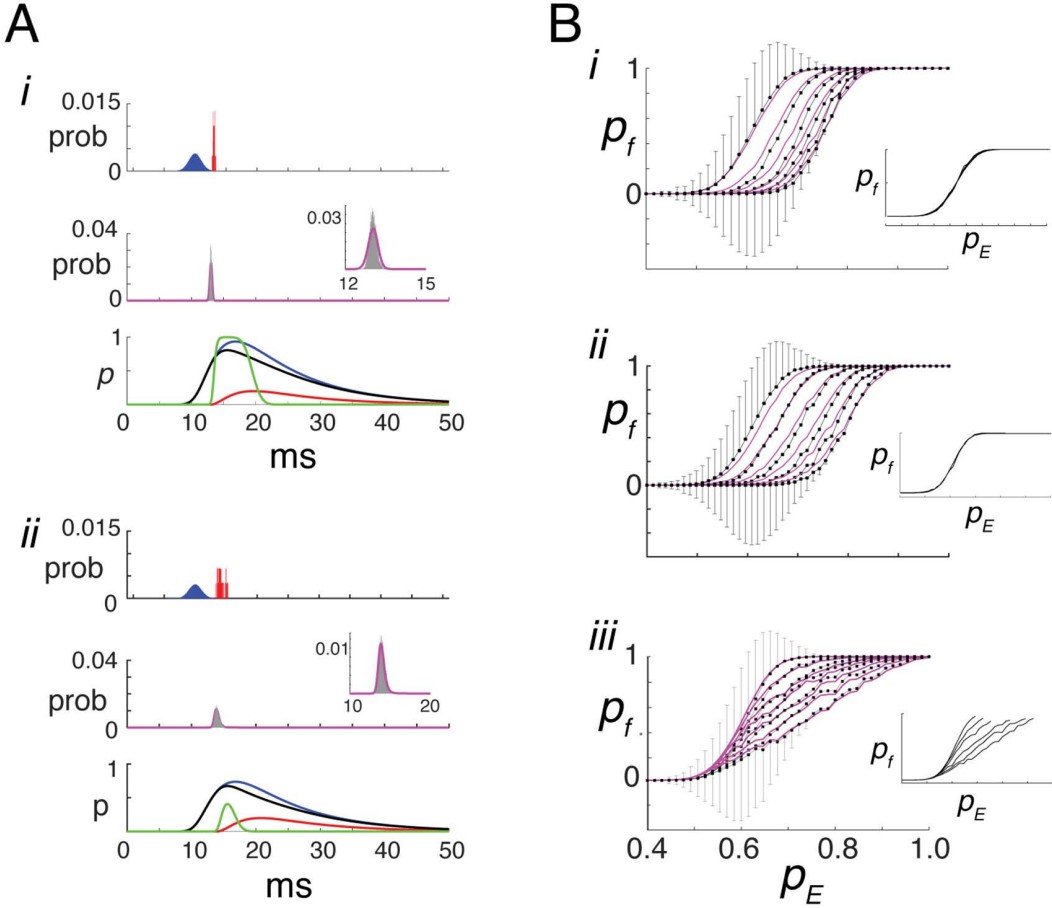

**Fig 6. Predicting firing probability for transient stimuli. A*i*,top**, probability distributions of EPSP (blue) and IPSP (red) arrival times in the reference cell. **Middle**, predicted firing probability (magenta) overlaid on the spike time histogram (gray); **Inset**, magnified view. **Bottom**, superimposed traces of $p_E \tilde{p}_E(t)$ (blue), $p_I \tilde{p}_I(t)$ (red), net excitatory drive $\tilde{p}_{net}(t)$ (black), and threshold-crossing probability (green). $p_E$ = 0.95, $k_n$ = 0.2, $k_a$ = 1, $k_{EI}$ = 1. **ii**, same but with $p_E$ = 0.75. **B,*i***, I-O curves showing mean firing probability (±SD) vs. $p_E$, for fixed $p_I$ values ($0 \leq p_I \leq 0.5$). The level was specified by adjusting and then fixing $k_n \in [0, 0.5]$. Other parameters are kept constant: $p_{E \rightarrow R}$ = 1, $p_{E \rightarrow I}$ = 0.8, $a_E = a_I = \pm 10.3$ pA, producing PSPs with amplitude $\pm 250\ \mu V$. Predicted values (magenta) are overlaid. Increasing $p_I$ shifts the curve rightward without changing slope. **Inset**, I-O curves aligned by threshold. **ii**, same but with $p_I$ increasing linearly with $p_E$, implemented by setting $k_n = k_c p_E$; $k_c \in [0, 1]$. **iii**, same but with $p_{E \rightarrow I} = k_p p_E$; $0.85 \leq k_p \leq 1$, $k_n$ = 1. **Model parameters:** $n_E$ = 100, $n_I = k_n n_E$. Histograms compiled over 5000 trials with bin width = 0.01 ms.

time-varying excitatory (blue) and inhibitory (red) probabilities thus follow the same temporal profiles as the compound EPSP and IPSP, respectively (Fig 6A*i*, bottom panel).

The time-dependent net excitatory drive—that is, the component of the excitatory input that is not canceled by inhibition (black curve in Fig 6A, bottom panel)—is defined as

$$\tilde{p}_{net}(t) = p_E \tilde{p}_E(t) - p_I \tilde{p}_I(t). \tag{7}$$

When excitation is present, this expression can be equivalently rewritten as

$$\tilde{p}_{net}(t) = p_E \tilde{p}_E(t) \left[ 1 - \frac{p_I \tilde{p}_I(t)}{p_E \tilde{p}_E(t)} \right],$$

where the ratio $\frac{p_I\,\tilde{p}_I(t)}{p_E\,\tilde{p}_E(t)}$ quantifies the instantaneous balance between inhibition and excitation. This time-dependent formulation is directly analogous to the sustained-stimulus case, where $p_{net} = p_E(1 - p_I)$ represents the steady-state probability that excitation survives inhibition.

The difference form provides the primary definition of $\tilde{p}_{net}(t)$ and remains well defined throughout the stimulus, including times at which excitation vanishes. To preserve its interpretation as a probability of excitation, negative values are rectified to zero, ensuring $\tilde{p}_{net}(t) \geq 0$ for all $t$.

The timing of action potentials is computed in two stages (see S1 Appendix). First, the probability that the net excitatory input exceeded the firing threshold was computed from $\tilde{p}_{net}(t)$ (bottom, green curve; Eq S29 of S1 Appendix). Second, this probability was used to derive the probability of firing at each time (Eq S30 of S1 Appendix), accounting for the constraint that a neuron can fire only once and only if it has not already fired. This procedure yields the predicted firing probability (middle panel, magenta curve), which closely matches the spike histogram (gray, inset). Similar results were obtained for a weaker stimulus (Fig 6A*ii*) and when the synaptic inputs were delivered as conductances (Fig S4 of S1 Appendix).

Modulating inhibitory strength led to systematic shifts in the I–O curves. The total probability of firing $p_f$ was calculated either from the area under the spiking histogram (gray) or from the predicted probability (magenta). Fig 6B*i* shows $p_f$ plotted against $p_E$ for different levels of $p_I$, analogous to the green curves in Fig 1B. The value of $p_I$ was controlled by adjusting and then fixing $k_n$ in Eq 3 so that $p_I$ remained constant across values of $p_E$. As $p_I$ increased, the level of required to evoke firing also increased, resulting in a rightward shift of the I–O curves without a change in slope. When aligned by their respective thresholds, the curves collapsed onto a single profile (inset). Thus, unlike in the toy model or under sustained input conditions, the effect on the I–O relationship was purely additive.

Similar results were obtained when $p_I$ increased linearly with $p_E$ (as in Fig 3C). This was implemented by letting $k_n$ increase as $k_n = k_c p_E$. As the slope of $p_I$ vs $p_E$ increased, the curves shifted to the right (Fig 6B*ii*), but unlike the toy model (Fig 1B, orange curves) and tonic input (Fig 3C), remained monotonic with no change in slope (inset). Thus, the effect on the I–O curve was also additive.

In Fig 6B*iii*, the inhibitory input probability $p_I$ was modulated by varying $p_{E \to I}$ so that, as in the sustained-stimulus condition (Fig 3D, inset), the resulting $p_I$ reflected the I–O function of the inhibitory neurons. As $p_{E \to I}$ increased, the slopes of the I–O curves of the reference cell decreased, whereas their thresholds remained constant. Unlike in the simple model, the curves remained monotonically increasing, with those obtained under weaker inhibition saturating at high $p_E$. Thus, under these conditions, the effect on the I–O curve was predominantly multiplicative within the range before saturation (inset).

Taken together, these results show that for transient inputs the effect of inhibition on firing probability depends on how inhibitory recruitment interacts with threshold crossing during the transient response (see S1 Appendix for a detailed analysis). When inhibitory strength was held constant or scaled proportionally with excitation (Fig 6B*i–ii*), the I-O curves shifted to higher $p_E$ values without a change in slope. In contrast, when inhibitory recruitment increased with excitation through the feedforward pathway (Fig 6B*iii*), the sensitivity of firing probability to $p_E$ changed, producing multiplicative gain modulation.

## Discussion

This study aimed to elucidate the principles governing stimulus-driven excitatory–inhibitory (E–I) interactions in feedforward inhibitory circuits. By combining excitatory and inhibitory effects within a probabilistic framework, a simple relationship was derived that predicts how E–I balance shifts during stimulation. This formulation links those shifts to both cellular and network-level variables, offering mechanistic insight into how inhibition shapes neuronal I–O functions, modulates gain, and influences overall response dynamics.

A probabilistic formulation offers several theoretical advantages over classical approaches that rely on summing synaptic currents to predict firing [51]. First, the multiplicative expression introduced here captures the survival of excitatory inputs

under feedforward inhibition in a form that is both biologically interpretable and analytically tractable. Because probabilities are inherently bounded between 0 and 1, the resulting I–O curves are automatically rectified and sigmoidal—features that are often imposed manually in models to prevent negative or unbounded firing probabilities or rates [10,19,30,62–64].

Second, the net probability $p_{net}$ can be used directly in analytic expressions to derive I–O relationships and characterize gain control. The reduced form naturally incorporates the variables that determine $p_I$ (such as interneuron recruitment and synaptic reliability) and automatically scales with $p_E$, eliminating the need for manual adjustment when analyzing I–O curves. Because $p_{net}$ closely tracks temporal structure of firing patterns, it can be used directly, or as input to Poisson point-process models [65], to efficiently predict stimulus-evoked responses with realistic temporal statistics.

Third, this framework explicitly incorporates synaptic noise, yielding smoothly rising I–O curves without requiring additional assumptions. The sources of variability are well defined and can, in principle, be used to compute statistical properties of both inputs and outputs, linking fluctuations in synaptic drive to the variability and reliability observed in neuronal firing.

### Relation to previous work

The model's I–O curves exhibit two distinct regimes. For weak stimuli (low $p_E$), firing occurs in a sub-oscillatory regime driven primarily by input variability, producing a sigmoidal rise. This behavior aligns with observations in visual cortex, where input summation is supralinear for weak stimuli and sublinear for strong ones [64,66]. The underlying mechanisms may differ, however, since the cortical responses reflect recurrent connectivity, which is absent in the present feedforward model. As input strength increases, the system transitions to an oscillatory regime, in which the I–O curve may continue to rise, flatten, or become non-monotonic. The specific shape of the curve is determined by how inhibition scales with excitation, consistent with previous modeling results [19].

The model highlights key variables that govern the transformation of synaptic input into spiking output. Understanding this transformation has practical implications, as it may enable the inference of intracellular dynamics from extracellular recordings [67]. Although direct inversion of the model equations is not feasible due to the number of interacting variables (e.g., $p_E$, $p_I$, $k_n$, $k_q$, $k_r$), the derived relationships can nevertheless inform which parameters must be independently measured or constrained to obtain reliable estimates.

### Gain modulation

Multiplicative gain modulation of I–O curves is essential for maintaining tuning acuity across different cognitive or behavioral states [27,28,68]. The present model reveals multiple mechanisms for achieving such modulation by varying how $p_I$ scales with $p_E$ (Fig 3B–3D). Unlike previous models [10,13,30], these effects arise without invoking recurrent connectivity, synaptic noise, or conductance-based mechanisms. Moreover, the framework provides a means to predict neuromodulatory influences based on their actions on synaptic or biophysical parameters [69,70].

For brief stimuli, in which firing is strongly influenced by the delay between excitation and inhibition [4,7,59], multiplicative gain emerged only when the drive to inhibitory neurons increased with $p_E$ (Fig 6B*iii*)—a condition likely to occur with natural stimuli. Optogenetic activation of inhibitory neurons during brief sensory stimulation [33] most closely resembles the simulations with fixed $p_I$ (Fig 6B*i*), which predict additive modulation of the I–O curves. The mixed multiplicative and additive effects observed experimentally may arise because optogenetic stimulation recruits inhibitory neurons in a manner that is partially decoupled from their natural, stimulus-dependent activation, placing the system in an intermediate regime between the scenarios considered here. For prolonged stimuli, the model predicts that modulation is predominantly multiplicative, consistent with observations during sustained optogenetic activation of inhibitory neurons [34].

More broadly, these results highlight a fundamental distinction between gain control under transient and sustained stimulation. For sustained inputs, excitation and inhibition reach a steady-state balance, so changes in inhibitory strength directly rescale the effective drive and naturally give rise to divisive gain modulation. For transient inputs, however, firing is

dominated by threshold crossing within a brief temporal window, and inhibition can influence firing either by delaying this crossing—producing additive shifts of the I–O relation—or, when inhibitory recruitment itself scales with excitatory drive, by altering the sensitivity of firing probability to further increases in $p_E$, resulting in true multiplicative gain modulation.

### Temporal response patterns and non-monotonic input–output relations

The model reproduced many of the temporal firing patterns observed in cortical and subcortical regions [35–39,46,53] by adjusting the magnitude and timing of the excitatory and inhibitory probabilities (Fig 5). A key prediction is that transient firing and non-monotonic I–O curves observed experimentally [16,35] need not arise from disproportionately strong inhibition. Both phenomena can emerge even when $p_E$ and $p_I$ increase in a balanced or proportional manner (Fig 2B, Fig 3C–3D, Fig 5).

The model further predicts that onset–offset responses can occur when both excitatory and inhibitory probabilities approach unity, with small shifts in inhibitory delay producing distinct temporal response components (Fig 5; [42]). Whether similar mechanisms operate in cortex remains unclear, as onset and offset responses may be inherited from subcortical pathways [40,53] or generated *de novo* within cortical circuits [22,41,44].

### Limitations

The theory relies on several key assumptions. First, the calculation of $p_{\text{net}}$ assumes that inhibitory inputs sum linearly with, and effectively cancel, excitatory inputs. Although the sublinear summation associated with conductance changes can be minimized through a correction term, the model does not account for heterogeneity in the amplitudes of individual EPSPs and IPSPs. For analytical simplicity, IPSP amplitudes were expressed as fixed fractions of the corresponding EPSPs. In reality, postsynaptic potential size depends on several factors, including synaptic location and the identity of the presynaptic population. For example, thalamocortical synapses are typically stronger than cortico-cortical synapses [71]. Moreover, IPSPs at the soma can, in principle, shunt or cancel multiple small EPSPs originating in distal dendrites.

Second, the constants ($k_n, k_{EI}, k_q, k_r$) that scale $p_I$ were chosen so that their product lies between 0 and 1, ensuring that $p_I$ remains interpretable as a probability. Although precise values for these constants are unlikely to be fully known in any given system, reasonable estimates can be drawn from the literature ($k_n$, [72,73]; $k_a, k_q$, [8,9,74]; $k_r$, [8,9,74]; $k_{EI}$, [49,57,75]). In addition, $k_q$ or $k_a$ may vary dynamically, reflecting short-term plasticity of excitatory and inhibitory synapses [9,55]. For modeling purposes, parameters should be constrained so that the product of these constants remains below unity.

Third, the model assumes that recurrent connections do not substantially contribute to the evoked firing rate [76]. In the cortex, the connection probability between excitatory neurons is relatively low [9,55]. Consequently, if the active region is spatially restricted—as in tuned responses driven by topographically organized inputs—the small number of recruited local excitatory neurons would exert minimal impact [76]. Nonetheless, incorporating recurrent connectivity will be important for extending the model to more general network configurations.

Fourth, spontaneous activity—whether intrinsically generated or originating from neurons outside the feedforward pathway—is not explicitly included in the present analysis. Background activity can alter the initial conditions of the network, influence transient responses [77,78], and contribute to steady-state firing. In principle, such effects could be incorporated into the probabilistic framework if expressed in terms of effective input probabilities, although this would require additional theoretical development.

Finally, a potential limitation of the framework is that spiking is described as being determined solely by inputs within the current time bin, without explicit dependence on prior activity. This simplification is partly mitigated by the model's structure. For brief stimuli, the probability of a spike at a given time is defined conditionally on no spikes having occurred in earlier bins, thereby incorporating past spiking history implicitly into the first-spike probability. In addition, the time-dependent changes in probability follow the shape of the underlying synaptic potentials, which reflect recent input

dynamics. For long-duration stimuli, the analysis focuses on the average synaptic current, which depends primarily on the overall rate and distribution of synaptic events. In this regime, the precise temporal sequence of individual inputs becomes less critical, as the mean current is determined by their statistical properties.

Despite these limitations, the model effectively captures many key aspects of stimulus-evoked responses and can serve as a foundation for developing formal mathematical analyses [79] to study more complex network configurations.

## Materials and methods

### Network parameters

The circuit consists of a reference neuron, whose firing activity serves as the model's output, and 20–50 local inhibitory neurons. Both cell types received $n_E$ afferents from an excitatory source, with the reference cell also receiving inputs from $n_I$ interneurons. Typical values of the variables used in the simulations are listed in Table 2.

### Leaky integrate-and-fire parameters

Simulations were performed in the MATLAB programming environment. Neurons were modeled as standard leaky integrate-and-fire (LIF) units governed by

$$C\frac{dV}{dt} = -\frac{V - V_I}{R} + i_E + i_I,$$

(8)

where $R$ = 75 MΩ, $V_I$ = –70 mV is the resting potential, $\tau_m$ = 10 ms is the membrane time constant, $C$ = 133 pF is the capacitance, and the action potential threshold was set to –55 mV. After an action potential, the membrane potential was reset to $V_I$. The terms $i_E$ and $i_I$ denote the total excitatory and inhibitory synaptic currents, respectively (in nA). The integration time step (bin width) was $\Delta t$ = 0.01 ms.

Each unitary synaptic current $u_x(t)$, evoked by a single presynaptic spike, was described by a scaled alpha function:

**Table 2. Typical network parameter values used in simulations.**

| Parameter | Definition | Typical value (range) |
| --- | --- | --- |
| $n_E$ | Number of excitatory afferents | 100 or 250 |
| $n_I$ | Number of inhibitory neurons | 20 or 50 |
| $a_E, a_I$ | Amplitudes of excitatory and inhibitory synaptic currents | 10.3 and –10.3 pA |
| $q_E, q_I$ | Charge transfer of afferent and inhibitory inputs | 0.0557 pC |
| $r_E$ | Firing rate of afferents | 50 Hz |
| $r_I$ | Firing rate of inhibitory neurons | varied (50–200 Hz) |
| $k_n$ | Ratio of inhibitory to excitatory inputs ($n_I/n_E$) | varied (0–1) |
| $k_q$ | Ratio of unitary inhibitory to excitatory charge ($|q_I/q_E|$) | 1 |
| $k_r$ | Ratio of inhibitory to excitatory firing rates ($r_I/r_E$) | varied (1–4) |
| $k_{EI}$ | Relative synaptic reliability (EPSP vs IPSP) | 1 |
| $p_E$ | Prob. afferent fires given stimulus | varied (0–1) |
| $p_I$ | Effective prob. that an inhibitory spike appears in the reference cell | varied (0–1) |
| $p_{E\rightarrow R}$ | Prob. EPSP in reference cell given afferent spike | 1 |
| $p_{E\rightarrow I}$ | Prob. EPSP in inhibitory cell given afferent spike | varied (0–1) |
| $p_{I\rightarrow R}$ | Prob. IPSP in reference cell given inhibitory spike | 1 |

$$u_x(t) = a_x\, u_{\text{PSC}}(t),$$

$$u_{\text{PSC}}(t) = \frac{t}{\tau}\, e^{(1-t/\tau)}, \quad x \in \{E, I\},$$

(9)

where $\tau = 2$ ms and the peak amplitude was normalized to 1. For current-based simulations, $a_E = 10.3$ pA and $a_I = -10.3$ pA. For conductance-based simulations, $a_E = 0.147$ nS and $a_I = 1.045$ nS, with reversal potentials $V_E = 0$ mV and $V_I = -80$ mV. Under these conditions, the corresponding excitatory and inhibitory postsynaptic potentials (PSPs) had amplitudes of approximately $+250\ \mu$V and $-250\ \mu$V, respectively, when measured at the resting potential.

## Stimulus parameters

**Sustained stimulus.** For sustained stimulation, excitatory synaptic barrages to the reference and inhibitory neurons were generated by creating a spike train according to a Poisson process with total rate $\lambda = n_E p_E r_E$, where $r_E = 50$ Hz is the firing rate of a single afferent and $n_E = 250$ is the number of afferents. The excitatory barrage to the inhibitory cells was adjusted by scaling the total rate by $p_{E\to I}$. The excitatory barrage (0.1–1 s duration) was then obtained by convolving the spike trains with $u_E$.

To construct the inhibitory barrage, the excitatory barrages were delivered to $3n_I\ (= 3k_n n_E)$ inhibitory cells, and the resulting spike trains were stored in a matrix. From this set, approximately $p_I n_I$ trains were randomly selected, summed, and subsequently convolved with $u_I$. This inhibitory barrage was then delivered together with the excitatory barrage to the reference neuron. The average firing rate and peristimulus time histograms of the reference neuron were computed over 100–10000 trials, with each trial using different realizations of the synaptic barrages.

To examine gain modulation of I-O functions (Fig 4), the excitatory input probability was modeled as a Gaussian function,

$$p_E(x) = a\, e^{-\frac{(x - x_{\text{center}})^2}{2\sigma_x^2}},$$

where $p_E(x)$ denotes the *effective* excitation probability at stimulus feature $x$, $x_{\text{center}}$ is the preferred stimulus, and $\sigma_x$ is the tuning width. This spatially tuned $p_E(x)$ was then used to generate the excitatory drive to both the reference cell and the inhibitory neurons, following the procedures described above. Average excitatory and inhibitory synaptic currents, together with the evoked firing rate, were computed over 100 trials and plotted as functions of $x$.

In a subset of simulations (Fig 5), the inhibitory population was bypassed. In these cases, the inhibitory input was generated using the same procedure as for the excitatory input and, after appropriate scaling, was delivered directly to the reference neuron. Both excitatory and inhibitory spike trains were modeled as inhomogeneous Poisson processes with time-varying probabilities, producing instantaneous rates

$$\lambda_{E\to R}(t) = p_E(t)\, r_E\, n_E,$$

$$\lambda_{I\to R}(t) = p_E(t)\, p_I(t)\, r_I\, n_I.$$

(10)

The functions $p_E(t)$ and $p_I(t)$ shared the same temporal profile except for a fixed relative delay. Both ramped linearly from 0 to 1 over a 20 ms period, maintained the steady-state value for the duration of the stimulus, and then decayed back to zero. The inhibitory delay, defined as the onset difference between $p_E(t)$ and $p_I(t)$, was varied between $-2$ and $+2$ ms. Firing rates were set to $r_E = r_I = 50$ Hz, and the numbers of inputs were $n_E = n_I = 250$. Peristimulus time histograms (PSTHs) of the reference neuron were computed from 100–1000 independent trials.

**Transient stimuli.** In this mode, each afferent fired a single action potential in response to a brief stimulus. Temporal jitter was introduced so that the afferent spikes did not arrive synchronously across inputs. The distribution of excitatory

postsynaptic current (EPSC) arrival times to the reference and inhibitory neurons, denoted $h_E(t)$, was modeled as a Normal distribution scaled by $p_E n_E$:

$$h_E(t) = p_E n_E \mathcal{N}(\mu_t, \sigma_t^2), \qquad \mu_t = 5 \text{ ms}, \ \sigma_t = 1 \text{ ms},$$
$$h_{E \to R}(t) = p_{E \to R} h_E(t),$$
$$h_{E \to I}(t) = p_{E \to I} h_E(t). \tag{11}$$

Here, $h_{E \to R}(t)$ and $h_{E \to I}(t)$ represent the effective distributions of EPSC arrival times in the reference and inhibitory neurons, respectively, accounting for the fact that not all afferent spikes evoke synaptic currents unless $p_{E \to R}$ or $p_{E \to I}$ equal to 1.

Each histogram was convolved with the unitary synaptic kernel $a_E u_{\text{PSC}}(t)$, which defines the time course of a single postsynaptic current, to obtain the compound excitatory current in the reference and inhibitory neurons. Independent realizations of $h_E(t)$ were generated for each inhibitory cell, and their evoked spike times were documented.

A corresponding histogram of inhibitory spike times, $h_I(t)$, was compiled from the responses of the inhibitory neurons. This histogram was scaled by $p_{I \to R}$ and convolved with $a_I u_{\text{PSC}}(t)$, which defines the time course of a unitary inhibitory synaptic current. Because inhibitory activity was estimated conditionally on stimulus-evoked inhibition (see S1 Appendix), the resulting current was delivered directly to the reference neuron together with the excitatory current. Each simulation was repeated 1000–5000 times to compile the peristimulus spike histogram of the reference neuron (Fig 6A, middle panel, gray).

Inhibitory timing was estimated from trials in which inhibitory spiking occurred (i.e., conditioning on stimulus-evoked inhibition), yielding $\mathcal{P}_I(t \mid \text{stim})$ and $\tilde{p}_I(t \mid \text{stim})$; consequently, no additional factor of $p_E$ is applied.

Methods for calculating the time-dependent probabilities and predicted spike times are provided in S1 Appendix.

## Supporting information

**S1 Fig. Model parameters.**
(TIF)

**S2 Fig. Effects of conductance.**
(TIF)

**S3 Fig. Effects of large tuned input.**
(TIF)

**S4 Fig. Transient response in conductance mode.**
(TIF)

**S1 Appendix. Derivation of equations.**
(PDF)

## Author contributions

**Conceptualization:** Alex D. Reyes.

**Data curation:** Alex D. Reyes.

**Formal analysis:** Alex D. Reyes.

**Funding acquisition:** Alex D. Reyes.

**Investigation:** Alex D. Reyes.

**Methodology:** Alex D. Reyes.

**Project administration:** Alex D. Reyes.

**Resources:** Alex D. Reyes.

**Software:** Alex D. Reyes.

**Supervision:** Alex D. Reyes.

**Validation:** Alex D. Reyes.

**Visualization:** Alex D. Reyes.

**Writing – original draft:** Alex D. Reyes.

**Writing – review & editing:** Alex D. Reyes.

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
