## [Decision Letter · Decision Letter 0]

9 Aug 2025

Computing the effects of excitatory-inhibitory balance on neuronal input-output properties

PLOS Computational Biology

Dear Dr. Reyes,

Thank you for submitting your manuscript to PLOS Computational Biology. After careful consideration, we feel that it has merit but does not fully meet PLOS Computational Biology's publication criteria as it currently stands. Therefore, we invite you to submit a revised version of the manuscript that addresses the points raised during the review process.

Please submit your revised manuscript within 60 days Oct 09 2025 11:59PM. If you will need more time than this to complete your revisions, please reply to this message or contact the journal office at ploscompbiol@plos.org. Please include the following items when submitting your revised manuscript:

We look forward to receiving your revised manuscript.

Kind regards,

Sacha Jennifer van Albada

Academic Editor

PLOS Computational Biology

Hugues Berry

Section Editor

PLOS Computational Biology

**Journal Requirements:**

3) Please amend your detailed Financial Disclosure statement. This is published with the article. It must therefore be completed in full sentences and contain the exact wording you wish to be published.

2) If any authors received a salary from any of your funders, please state which authors and which funders..

4) Thank you for stating 'The source code and data used to produce the results and analyses presented in this manuscript are available on a Github repository at https://github.com/AlexDReyes/ReyesPlos_2025' Please note that, though access restrictions are acceptable now, your entire minimal dataset will need to be made freely accessible if your manuscript is accepted for publication. This policy applies to all data except where public deposition would breach compliance with the protocol approved by your research ethics board. If you are unable to adhere to our open data policy, please kindly revise your statement to explain your reasoning and we will seek the editor's input on an exemption.

5) Kindly revise your competing statement to align with the journal's style guidelines: 'The authors declare that there are no competing interests.'

**Reviewers' comments:**

Reviewer's Responses to Questions

**Comments to the Authors:**

Reviewer #1: This article provides new and interesting insights into the role of excitatory-inhibitory balance upon the key feature of sensory-driven processes. The results are sound, but the explanation of the model presented here is somewhat unclear and could be substantially improved, as outlined below.

Previous work by the author lays a very good foundation for simulations and analysis of EI-balanced neurons that is carried out in the current paper. A number of simplifications are made in the model, including only a single population of excitatory and inhibitory (resp.) neurons being included (i.e., the diversity of neurons, particularly inhibitory neurons, in cortical circuits is not considered, the synaptic time constants of inhibitory and excitatory neurons are identical (2 ms), etc.). However, these are no unreasonable approximations in a study of this sort that seeks to provide new insights into neural mechanisms, since they would not fundamentally alter the key properties, but rather represent fine-tuning or modulation of the effects.

The neuronal model used in this paper is based upon the well-known integrate-and-fire neuron (IAF) model with current-based synapses, which has been extended in this study to include an additional parameter that describes the probability of a postsynaptic response being initiated once an action potential arrives at the synapse. This modification of the standard IAF mechanism is reasonable, but features of it require additional discussion and/or justification based upon experimental evidence and values measured. The question of what changes could be expected when conductance synapses are used (rather than the current-injection synapses used here) is addressed well. One omission is a discussion of the role of spontaneous activity, which impacts the latency-to-first-spike in IAF neurons – the lack of spontaneous activity means that the voltage starts at the reset value for transient stimuli. A number of other approximations and potential shortcomings of the study, including the omission of lateral (recurrent) inputs, are acknowledged and briefly discussed. The figures generally provide good illustrations of the concepts and summary of the key results.

This study builds upon previous work by the author and other researchers, which is generally well referenced. However, one omission appears to be the substantial body of related work on EI-balanced networks by Deneve and colleagues (refs below), which seems to be very relevant to the current study (if there are grounds for this not being the case, then this also requires explanation).

Detailed comments

As mentioned above, a number of details of the presentation of the model are currently unclear, which makes it difficult for readers to readily interpret the results. The methods could be substantially improved, including as follows:

• There is a confusing mixture of Results and Methods (model) in the current manuscript. It is not possible to understand the Results in any detail without first having a clear explanation of the model. This partly explains why the first part of the Results section contains large amounts of text that rightly belongs in the Methods. However, this leads to considerable confusion for the reader, which could be overcome by a clear description of the Methods before the Results are presented.

• Throughout, there needs to be a clear notational difference between the average, instantaneous, and integral values of parameters, which otherwise leads to possible confusion.

• Figure 2: The caption is missing the details for subplots B(ii) and B(iii). It is also unclear what the bin-width is for the spike histograms. Some of the described steady-state values in the caption do not agree with those plotted.

• Appendix: The Equation-numbers and Figure-numbers in the appendices need to be distinguished from those in the main text (this is most simply done by adding an “S” at the start of each number). Otherwise, there is confusion about which equation or figure is being referred to.

• Appendix: The Figures in the appendix should be given subplot labels similar to that given to subplots in the main text.

• Appendix “Correcting for conductance” & Figure S1: Equation(S1) needs more information: what is “coul” and “sec”? How is G_E defined? Confusing notation to have g_E(t) and g_E(v) – one of these needs to be different (i.e., $\hat{g}_{E}(v)$). How is G_E defined (i.e., what are the limits on the time-integral)?

• Appendix “Effects of conductance for brief stimuli” & Figure S3: Details of the parameters are currently missing. The parameter details need to be sufficient for the plots to be reproduced.

Overall, though, these are relatively minor issues whose resolution would assist readers. The results and analysis presented in the paper make an important contribution to our understanding of the role of excitatory-inhibitory balance upon the key feature of sensory-driven processes.

References:

• Boerlin, M., Machens, C. K., and Deneve, S. (2013). Predictive Coding of Dynamical Variables in Balanced Spiking Networks. PLOS Computational Biology, 9(11):e1003258.

• Gutierrez, G. J. and Deneve, S. (2019). Population adaptation in efficient balanced networks. eLife, 8:e46926.

• Brendel, W., Bourdoukan, R., Vertechi, P., Machens, C. K., and Den`eve, S. (2020). Learning

to represent signals spike by spike. PLOS Computational Biology, 16(3):e1007692.

• Zeldenrust, F., Gutkin, B., and Deneve, S. (2021). Efficient and robust coding in heterogeneous recurrent networks. PLOS Computational Biology, 17(4):e1008673.

Reviewer #2: The review is uploaded as an attachment.

Reviewer #3: In the manuscript, the author introduces and studies a simple model to understand the operation of a standard cortical 'motif', that is, a feed-forward inhibitory circuit.

In this circuit, an excitatory neuron receives direct excitatory inputs as well as, possibly delayed, inhibitory inputs driven by the same excitatory inputs. The 'putative' operation of this circuit has been extensively investigated in sensory cortices.

The author shows that many experimentally-observed features of evoked neuronal responses (e.g., modulation of the f-I curve) can be qualitatively understood within this model just by taking into account (i) the probabilistic nature of synaptic integration/transmission and (ii) the 'level' of excitatory-inhibitory balance.

I found the manuscript quite interesting and, in particular, its perspective especially refreshing. There is an increasing emphasis in achieving quantitatively accurate predictions (which is important, to be sure), that one almost forgets the importance of qualitative understanding in terms of simple ('interpretable' is the modern term, I think) models, such as the one presented here.

I have just a few (i.e., 2) very minor comments that essentially amounts to some clarification.

The probabilities, p_E and p_I, are probabilities per unit time or alternatively, I'm thinking of time in a discretized way. What is somehow important to clarify, I think, is that, in the model, I'm thinking of neural integration (and hence spike generation) as a memory-less process, i.e., I can determine the probability of spiking (in this time bin, or per unit time) just looking at current inputs. There are situations, that have been extensively studied (e.g., escape noise or spike-response model), where this is indeed justified; but not always. This should be somehow shortly discussed.

This is just a matter of taste and the author is free to ignore this comment.

Why introducing 3 different symbols, k_n, k_q, k_r, in Eq. 2 when they all occur as a product? Also this entails, it seems to me, a problem in justifying why this product is expected to be smaller than 1 (see the discussion in Limitations, which, by the way, is not fully convincing). One could just by-pass all of this by introducing, one k, stipulate that it is smaller than 1 (by definition) and then discuss which 'real-world' variables (e.g., relative proportion of excitatory and inhibitory neurons) are supposed to affect its value and how.

**Have the authors made all data and (if applicable) computational code underlying the findings in their manuscript fully available?**

Reviewer #1: **No:** The github link provided gives a 404 error message

Reviewer #2: **No:**

Reviewer #3: None

PLOS authors have the option to publish the peer review history of their article (what does this mean? ). If published, this will include your full peer review and any attached files.

**Do you want your identity to be public for this peer review?** For information about this choice, including consent withdrawal, please see our Privacy Policy .

Reviewer #1: **Yes:** Anthony N. Burkitt

Reviewer #2: No

Reviewer #3: No

**Figure resubmission:**

**Reproducibility:**



---

## [Decision Letter · Decision Letter 1]

29 Jan 2026

Dear Dr. Reyes,

We are pleased to inform you that your manuscript 'Computing the effects of excitatory-inhibitory balance on neuronal input-output properties' has been provisionally accepted for publication in PLOS Computational Biology.

Best regards,

Sacha Jennifer van Albada

Academic Editor

PLOS Computational Biology

Hugues Berry

Section Editor

PLOS Computational Biology

Reviewer's Responses to Questions

**Comments to the Authors:**

Reviewer #1: The author has addressed all the concerns raised in the original review.

Reviewer #3: The author has addressed satisfactorily my few, rather minor, comments.

**Have the authors made all data and (if applicable) computational code underlying the findings in their manuscript fully available?**

Reviewer #1: Yes

Reviewer #3: Yes

PLOS authors have the option to publish the peer review history of their article (what does this mean? ). If published, this will include your full peer review and any attached files.

**Do you want your identity to be public for this peer review?** For information about this choice, including consent withdrawal, please see our Privacy Policy .

Reviewer #1: **Yes:** Anthony N. Burkitt

Reviewer #3: No

---

## [Editor Report · Acceptance letter]

PCOMPBIOL-D-25-00980R1

Computing the effects of excitatory-inhibitory balance on neuronal input-output properties

Dear Dr Reyes,

I am pleased to inform you that your manuscript has been formally accepted for publication in PLOS Computational Biology. Your manuscript is now with our production department and you will be notified of the publication date in due course.

With kind regards,

Lilla Horvath
